# Scent of Knowledge: Optimizing Search-Enhanced Reasoning with Information Foraging

**Hongjin Qian**[1], **Zheng Liu**[1,2*]

[1] Beijing Academy of Artificial Intelligence
[2] Hong Kong Polytechnic University
{chienqhj,zhengliu1026}@gmail.com

## Abstract

Augmenting large language models (LLMs) with external retrieval has become a standard method to address their inherent knowledge cutoff limitations. However, traditional retrieval-augmented generation methods employ static, pre-inference retrieval strategies, making them inadequate for complex tasks involving ambiguous, multi-step, or evolving information needs. Recent advances in test-time scaling techniques have demonstrated significant potential in enabling LLMs to dynamically interact with external tools, motivating the shift toward adaptive inference-time retrieval. Inspired by Information Foraging Theory (IFT), we propose InForage, a reinforcement learning framework that formalizes retrieval-augmented reasoning as a dynamic information-seeking process. Unlike existing approaches, InForage explicitly rewards intermediate retrieval quality, encouraging LLMs to iteratively gather and integrate information through adaptive search behaviors. To facilitate training, we construct a human-guided dataset capturing iterative search and reasoning trajectories for complex, real-world web tasks. Extensive evaluations across general question answering, multi-hop reasoning tasks, and a newly developed real-time web QA dataset demonstrate InForage's superior performance over baseline methods. These results highlight InForage's effectiveness in building robust, adaptive, and efficient reasoning agents. We provide all codes and datasets in the supplementary materials as well as in *this repository*.

## 1 Introduction

Augmenting large language models (LLMs) with external knowledge retrieved via search tools is a common approach to address their inherent knowledge cutoff limitation [Lewis et al., 2020a, Zhao et al., 2024a]. Existing methods typically apply a static, pre-inference retrieval strategy by concatenating retrieved information into the input prompt, enabling the LLM to generate answers based on the provided external knowledge [Gao et al., 2024]. However, this approach often lacks adaptiveness, especially for complex tasks in which users' information needs are ambiguous, rationale-based, or not directly searchable, as these tasks require iterative reasoning to progressively uncover the necessary evidence for answer generation [Qian et al., 2025a].

Recent advances in test-time scaling techniques have significantly strengthened LLMs' reasoning abilities, enabling complex behaviors such as long chain-of-thought reasoning and dynamic tool use [Snell et al., 2024, Muennighoff et al., 2025]. Reasoning-based models such as OpenAI's o1 and DeepSeek R1 have demonstrated significant gains on challenging tasks, particularly in areas like mathematical problem-solving and coding, where iterative self-reflection and refinement are essential for success [OpenAI, 2024, DeepSeek-AI, 2025]. Inspired by these developments, a natural extension

---

[*]Corresponding author.

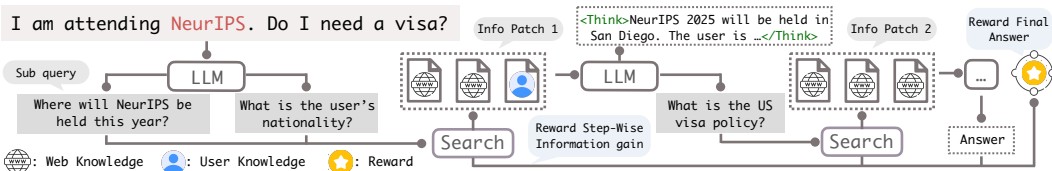

Figure 1: Illustration of InForage. Complex tasks often require multi-hop reasoning and iterative retrieval across dispersed knowledge sources. InForage integrates retrieval into each reasoning step and assigns structured rewards based on the relevance and utility of each retrieved patch. This dynamic supervision encourages the model to progressively gather, evaluate, and integrate evidence—mirroring human-like information foraging.

for retrieval-augmented generation is to shift retrieval from a static, pre-inference step to a dynamic, inference-time process. This transition allows LLMs to iteratively retrieve and adaptively integrate external knowledge during reasoning, aligning more closely with the evolving and multi-layered information needs inherent to complex information seeking tasks [Li et al., 2025, Jin et al., 2025a].

However, effectively supporting such dynamic search-enhanced reasoning for complex tasks presents two key challenges. First, due to the implicit and evolving nature of such tasks, it is difficult to retrieve all necessary evidence through a single retrieval action. Each retrieval typically uncovers only a local *information patch*, which is a partial subset of the expected knowledge space, and it requires iterative accumulation and integration via reasoning to form a complete answer [Qian et al., 2025b]. Second, the value of an information patch is not intrinsic to the retrieved content itself but depends on its contribution to the overall reasoning process and final answer accuracy [Zhu et al., 2024, Zhou et al., 2024]. Therefore, each retrieval decision significantly influences not only the immediate reasoning step but also the ultimate correctness of the final response. To illustrate these challenges concretely, consider a simplified case shown in Figure 1: "I am attending NeurIPS. Do I need a visa?" Addressing this question requires navigating the vast information space by iteratively retrieving local *information patches* from distinct sources (e.g., external web pages and personal profiles). The system must first uncover the event's location before accurately resolving the user's actual visa requirement. These sequential dependencies, where each retrieval clarifies only part of the information need, highlight the importance of adaptive, iterative retrieval strategies capable of dynamically refining the reasoning trajectory.

While such information-seeking tasks remain challenging for current retrieval-augmented systems, humans often resolve them efficiently with just a few iterative online searches [Nakano et al., 2021, Shani et al., 2024]. This suggests that humans inherently possess adaptive strategies for navigating complex information spaces. Drawing inspiration from cognitive science, we turn to **Information Foraging Theory (IFT)** [Pirolli and Card, 1999], which provides a formal explanation for how humans strategically seek information by balancing the expected value gained from an *information patch* against the cognitive and time costs involved in exploring it. Central to IFT is the concept of *information scent*, defined as the perceived relevance or utility of available informational cues. Analogous to how animals follow scent trails to valuable resources, humans use information scent to guide their search toward promising information sources. Translating this analogy into retrieval-augmented reasoning, the model's intermediate reasoning steps and generated subqueries can be viewed as dynamically evolving assessments of information scent. Stronger information scent thus corresponds to higher-quality reasoning paths and subqueries, which lead retrieval actions toward more relevant and information patches. Consequently, optimizing intermediate information gain throughout the reasoning trajectory becomes as critical as ensuring the correctness of the final answer.

Building on this perspective, we propose **InForage**, a reinforcement learning (RL) framework designed to enhance the search-augmented reasoning capabilities of LLMs. Previous reasoning methods typically optimize models based solely on the correctness of final answers, overlooking the crucial role of intermediate retrieval steps. In contrast, InForage explicitly rewards effective information-seeking behavior at each stage of the reasoning process, recognizing that meaningful retrieval actions incrementally shape both the reasoning trajectory and the accuracy of the final response. Inspired by IFT, we introduce three complementary reward mechanisms to incentivize comprehensive reasoning behaviors: (1) an **Outcome Reward**, which credits trajectories leading to correct final answers; (2) an **Information Gain Reward**, which rewards intermediate retrieval

steps that uncover valuable evidence; and (3) an **Efficiency Penalty**, which discourages unnecessarily prolonged reasoning, encouraging concise and cost-effective information foraging.

Most existing QA datasets provide only final question–answer pairs, lacking records of intermediate reasoning or retrieval steps. Moreover, they often feature shallow queries that can be resolved with one or two retrievals, limiting their utility for training models on complex, multi-step reasoning. To address this gap, we construct a dataset that captures fine-grained human information-seeking trajectories through open-ended web browsing. Starting from a seed claim, annotators iteratively query search engines, select relevant documents, and extract rationale-dependent claims to formulate subsequent queries. After gathering sufficient evidence, we use a strong LLM (e.g., GPT-4o) to generate QA pairs that require multi-hop reasoning and layered evidence integration. Each example involves at least three information hops or intersecting conditions to ensure true complexity. Crucially, the dataset records every step of the search and reasoning process, enabling reward supervision over final answer correctness, intermediate retrieval quality, and overall reasoning efficiency.

We evaluate InForage on standard QA, multi-hop reasoning, and a self-constructed real-time web QA benchmark. Results show that InForage consistently outperforms baselines, validating the effectiveness of learning from richly supervised, search-augmented reasoning data.

In summary, the contributions of this paper are threefold: (1) We formalize search-enhanced reasoning through information foraging theory, modeling retrieval decisions as a dynamic optimization of information scent and patch value along the reasoning trajectory. (2) We propose InForage, a reinforcement learning framework that jointly optimizes outcome correctness, intermediate information gain, and reasoning efficiency, enabling LLMs to perform adaptive, multi-step information foraging. (3) We construct a human-guided search reasoning dataset that captures multi-step web browsing trajectories and rationale-dependent query formulation, providing the structured supervision necessary to train and evaluate InForage effectively.

## 2 Method

### 2.1 Preliminary

The information-seeking process using LLMs can be formulated as $\mathcal{Y} = \Theta(q)$, where $q$ is the input query, $\mathcal{Y}$ is the generated answer, and $\Theta(\cdot)$ denotes the LLM. Since the knowledge embedded in most LLMs is fixed after training and difficult to update, incorporating external information has become a common strategy to enhance their performance on information-seeking tasks. Retrieval-augmented generation (RAG) is a widely adopted approach that follows this paradigm. In RAG, given a query $q$, the system first retrieves relevant knowledge $\mathcal{K}$ from an external knowledge base $\mathcal{D}$ using a retriever $\Gamma(\cdot)$, and then generates the final answer conditioned on both $q$ and $\mathcal{K}$. The process can be defined as:

$$\mathcal{Y} = \Theta(q \mid \mathcal{K}), \quad \mathcal{K} = \Gamma(q \mid \mathcal{D}), \tag{1}$$

where $\Gamma(\cdot)$ represents the retrieval function and $\mathcal{D}$ is the external knowledge corpus.

However, in RAG systems, performance is tightly upper-bounded by retrieval quality: if the retrieved knowledge $\mathcal{K}$ is sub-optimal, the generated answer $\mathcal{Y}$ is likely to be flawed. This limitation becomes even more pronounced in complex knowledge discovery tasks, where information needs are implicit or multi-layered. In such cases, effective information gathering requires iterative reasoning and adaptive retrieval over multiple steps.

Recent advances in reasoning LLMs enable a more dynamic retrieval paradigm, shifting retrieval from a static pre-inference step to an adaptive, inference-time mechanism. Specifically, the generated output $\mathcal{Y}$ typically contains a *reasoning* segment and an *answering* segment, denoted as $(\mathcal{Y}_{\text{think}}, \mathcal{Y}_{\text{answer}}) \in \mathcal{Y}$. During reasoning, the model generates a trajectory of intermediate reasoning steps (or subqueries), exploring potential paths toward solving the task. Retrieval is interleaved into this process as:

$$\mathcal{Y} = \Theta\left(q \mid \mathcal{Y}_{\text{think}} \otimes \{\mathcal{K}_t\}_{t=1}^{T}\right), \quad \mathcal{K}_t = \Gamma(q_t^{\text{sub}} \mid \mathcal{D}), \quad q_t^{\text{sub}} \in \mathcal{Y}_{\text{think}}, \tag{2}$$

where $q_t^{\text{sub}}$ denotes the subqueries generated during reasoning, used to retrieve intermediate knowledge $\mathcal{K}_t$. The operator $\otimes$ indicates that retrieved knowledge is dynamically interleaved into the evolving reasoning trajectory.

## 2.2 Information Foraging Perspective on Search-Enhanced Reasoning

Suppose the expected knowledge space required for solving a complex information-seeking task consists of $n$ documents, denoted as $\mathcal{D}^* = [d_1, \ldots, d_n]$. Generating the correct answer necessitates recovering the complete set $\mathcal{D}^*$ in the reasoning process. At the $t$-th retrieval step, given the generated subquery $q_t^{\text{sub}}$, the model may retrieve only a partial *information patch* $\mathcal{K}_t \subseteq \mathcal{D}^*$, rather than the full set. Consequently, an ideal optimization objective is to sequentially gather all necessary information patches in as few reasoning-retrieval steps as possible, enabling accurate answer generation with minimal search effort.

Inspired by Information Foraging Theory, we characterize the search-enhanced reasoning process as *information foraging*, where the evolving *information scent* is expressed through the model's intermediate reasoning steps and generated subqueries. Guided by this scent, the model iteratively employs retrieval tools to gather local *information patches* from the broader knowledge space. A strong information scent—reflected in coherent reasoning trajectories and effective subqueries—facilitates the retrieval of increasingly relevant documents. Crucially, while accurate final answers depend on the successful accumulation of relevant information, even in cases where the final prediction is incorrect, a reasoning trajectory that effectively gathers valuable knowledge patches remains meaningful and should be explicitly rewarded. Formally, this process can be expressed as the following objective:

$$\max_{\{q_t^{\text{sub}}\}_{t=1}^T} \mathbb{E}\left(\mathcal{S}(\mathcal{Y}_{\text{answer}}) + \alpha \cdot \mathcal{C}\left(\bigcup_{t=1}^T \mathcal{K}_t, \mathcal{D}^*\right)\right) \cdot \beta^T, \tag{3}$$

where $\mathcal{S}(\mathcal{Y}_{\text{answer}})$ denotes the evaluation metric of the final generated answer, $\mathcal{C}\left(\bigcup_{t=1}^T \mathcal{K}_t, \mathcal{D}^*\right)$ measures the coverage of the retrieved patches relative to the expected knowledge $\mathcal{D}^*$, $T$ is the total number of reasoning-retrieval steps, and $\alpha, \beta \in (0, 1)$ are weighting factors controlling the trade-off between information completeness and trajectory efficiency.

## 2.3 The proposed method: InForage

Building on the information foraging perspective, we propose **InForage**, a search-enhanced reasoning method that enables LLMs to iteratively reason, retrieve, and integrate external knowledge for solving complex information-seeking tasks. The reasoning trajectory of InForage is structured into specialized stages and can be formalized as:

$$\mathcal{Y} = \underbrace{\texttt{<think>} \text{ reasoning content } \texttt{</think>} \texttt{<search>} \text{ subquery } \texttt{</search>}}_{\text{information scent}}$$

$$\underbrace{\texttt{<info>} \text{ retrieved information } \texttt{</info>}}_{\text{information patch}} \cdots \texttt{<answer>} \text{ final answer } \texttt{</answer>}. \tag{4}$$

As illustrated above, the model's generation process $\mathcal{Y}$ unfolds as a sequential composition of blocks: a `<think>` block capturing intermediate reasoning contents, a `<search>` block emitting subqueries, corresponding retrieval results encapsulated within `<info>` blocks, and finally, a `<answer>` block producing the final response. Throughout this process, the evolving *information scent*—expressed via reasoning contents and subqueries—guides the model to retrieve information patches that incrementally reconstruct the expected knowledge space, ultimately supporting accurate answer generation.

**Reward Design for InForage.** Reinforcement learning plays a key role in training reasoning models by enabling them to self-explore diverse reasoning trajectories and rewarding those that prove effective. This aligns model behavior with the objective of adaptive information seeking and efficient decision-making. However, since these trajectories are self-generated and do not always yield correct final answers, relying solely on outcome-based rewards leads to sparse supervision signals—making the training process harder to optimize and less sample-efficient [Chen et al., 2025]. Following the optimization goal defined in Eq. 3, we decompose the overall reward into three complementary components: **Outcome Reward**, **Information Gain Reward**, and **Efficiency Penalty**. Together, these rewards guide the model to reason more strategically, forage information effectively, and minimize unnecessary retrieval steps.

**(1) Outcome Reward:** At the end of each rollout, we extract the predicted answer $\mathcal{Y}_{\text{answer}}$ and evaluate it using a task-specific metric $\mathcal{S}(\mathcal{Y}_{\text{answer}})$, such as Exact Match or F1 score. The Outcome

Reward directly reflects the final task success:

$$R_{\text{outcome}} = \mathcal{S}(\mathcal{Y}_{\text{answer}}). \tag{5}$$

**(2) Information Gain Reward:** A strong information scent leads to the retrieval of more relevant documents, enhancing intermediate knowledge acquisition. To capture this behavior, we compute the cumulative coverage of the retrieved patches against the expected knowledge set $\mathcal{D}^*$. Specifically, at each retrieval step $t$, we evaluate the partial coverage $\mathcal{C}\left(\bigcup_{\tau=1}^{t} \mathcal{K}_\tau, \mathcal{D}^*\right)$ and define the Information Gain Reward as the maximum coverage achieved during the reasoning trajectory:

$$R_{\text{gain}} = \max_{t=1,\ldots,T} \mathcal{C}\left(\bigcup_{\tau=1}^{t} \mathcal{K}_\tau, \mathcal{D}^*\right). \tag{6}$$

**(3) Efficiency Penalty:** Since the minimal trajectory for search-enhanced reasoning involves at least two steps (one for searching and one for answering), we penalize excessively long reasoning paths. The Efficiency Penalty is defined as an exponential decay applied to the total reward:

$$R_{\text{efficiency}} = \beta^{\max(0, T-2)}, \quad \text{where} \quad 0 < \beta < 1. \tag{7}$$

**Final Reward:** The final reward assigned to a rollout aggregates the three components as:

$$R = R_{\text{efficiency}} \cdot \left(R_{\text{outcome}} + \alpha \cdot R_{\text{gain}}\right), \tag{8}$$

where $\alpha \in (0, 1)$ balances the emphasis between final answer correctness and intermediate information gathering.

**Optimization.** We first train the foundation LLMs using supervised fine-tuning (SFT) to enable the model to perform iterative reasoning and retrieval. Specifically, we construct the SFT dataset by providing gathered human web browsing trajectories and corresponding QA pairs to a strong LLM, which then generates step-wise reasoning and retrieval responses as the training target.

Subsequently, we optimize InForage using Proximal Policy Optimization (PPO) [Schulman et al., 2017], guided by the rule-based rewards defined previously. For each input prompt $p$, the model generates a trajectory $\mathcal{Y}$, interleaved with retrieved information patches $\mathcal{K}_{t\,t=1}^{T}$. At each generation step $t$, we calculate the reward $r_t$ and estimate the advantage $\mathcal{A}_t$ via Generalized Advantage Estimation (GAE) [Schulman et al., 2016]. Let $\mathcal{Y}_{<t}$ denote the prefix generated up to step $t$, and $\mathcal{K}_{<t}$ represent the retrieved information patches available thus far. The PPO training objective is defined as:

$$\mathcal{L}_{\text{PPO}}(\theta) = \mathbb{E}_t\left[\min\left(r_t \mathcal{A}_t,\, \text{clip}(r_t, 1 - \epsilon, 1 + \epsilon)\mathcal{A}_t\right)\right], \quad r_t = \frac{\pi_\theta(\mathcal{Y}_t \mid \mathcal{Y}_{<t}, \mathcal{K}_{<t})}{\pi_{\theta_{\text{old}}}(\mathcal{Y}_t \mid \mathcal{Y}_{<t}, \mathcal{K}_{<t})}, \tag{9}$$

where $\pi_\theta$ denotes the current policy, $\pi_{\theta_{\text{old}}}$ is the sampling policy from the previous iteration, and $\epsilon$ is the clipping parameter. Following Jin et al. [2025a], we restrict gradient updates exclusively to model-generated tokens, excluding tokens originating from retrieved information patches $\mathcal{K}$ by applying appropriate masking.

## 3 Experiments

### 3.1 Settings

**Baselines**: We compare InForage against both non-retrieval and retrieval-augmented baselines. For non-retrieval methods, we consider: (1) **Vanilla**, prompting the LLM to directly generate answers; (2) **SFT**, fine-tuning the LLM on the same QA pairs used by InForage; (3) **Reasoning**, training the LLM with reinforcement learning on QA pairs following DeepSeek-AI [2025]. For retrieval-augmented baselines, we include: (1) **Vanilla RAG**, which retrieves top-$k$ documents once and prepends them to the prompt; (2) **IRCoT** [Trivedi et al., 2022a], alternating retrieval with chain-of-thought reasoning; (3) **RQRAG** [Chan et al., 2024], which refines initial queries through rewriting and decomposition to improve retrieval accuracy; (4) **Self-RAG** [Asai et al., 2023], which introduces a self-reflection mechanism allowing the model to critique and revise its own outputs based on retrieved evidence; (5) **Search-o1** [Li et al., 2025], which enhances LLMs with an agentic retrieval module

and a Reason-in-Documents component for structured document reasoning; (6) **Search-R1** [Jin et al., 2025a], which learns to generate multiple search queries during reasoning via reinforcement learning to optimize multi-turn retrieval interactions.

**Datasets**: We evaluate on the following datasets: Natural Questions [Kwiatkowski et al., 2019], TriviaQA [Joshi et al., 2017], PopQA [Mallen et al., 2022], HotpotQA [Yang et al., 2018], 2WikiMultihopQA [Ho et al., 2020], MuSiQue [Trivedi et al., 2022b], Bamboogle [Press et al., 2023], and a self-constructed real-time web QA dataset. We report exact match (EM) as the primary metric.

### 3.2 Implementaion Details

We use Qwen-2.5 Instruct models (3B and 7B) as our foundation LLMs for InForage. For SFT, we first sample 4,000 question–answer pairs from our self-constructed training datasets. To generate corresponding reasoning trajectories, we condition Qwen-2.5-72B on each QA pair along with its associated relevant claims. These model-generated trajectories are then used to fine-tune the foundation models for two epochs using a learning rate of $1 \times 10^{-5}$. Following SFT, we perform RL with PPO over 300 steps, using a learning rate of $1 \times 10^{-6}$ and a warm-up ratio of 0.5. The RL training corpus includes data from our self-constructed dataset, Natural Questions (NQ), and HotpotQA. For NQ and HotpotQA, we use Wikipedia as the knowledge source; for the self-constructed dataset, we use the cached web pages gathered during its creation. Structured rewards (Eq. 8) with $\alpha = 0.2$ and $\beta = 0.95$ are applied only to the self-constructed dataset, as the others lack intermediate traces.

During inference, we use the E5 encoder [Wang et al., 2024] and the Wikipedia dump from FlashRAG [Jin et al., 2025b] as the retrieval backend for open-domain and multi-hop QA tasks. For complex, real-time web-based tasks, we cache Google Search results and scrape the full content of retrieved pages. Retrieval over this corpus is performed using the BGE-M3 retriever, and we set the maximum number of reasoning steps to 6.

Regarding baselines, we adopt reported results when available (e.g., Search-R1 [Jin et al., 2025a]) and reproduce results using official code and checkpoints when necessary. For RQRAG and Self-RAG, we evaluate their official 7B checkpoints using FlashRAG. All other baselines use Qwen2.5-3B-Instruct as the foundation model. For consistency, all retrieval-based methods use top-$k = 3$ retrieved documents. All training and evaluation were conducted using 8 NVIDIA A800-80G GPUs. We provide all source codes, datasets and prompts in *this repository*.

### 3.3 Training Dataset Construction

To effectively train InForage, we require structured supervision signals for intermediate information-gathering steps. However, most existing QA datasets provide only QA pairs without detailed records of the information-seeking process, making them unsuitable for this purpose. To address this gap, we construct a large-scale dataset designed explicitly to capture comprehensive human information-seeking trajectories, inspired by human information foraging behaviors.

**Open-Ended Information Browsing and Annotation:** To ensure that each example genuinely requires multi-step reasoning and cannot be answered via simple direct search, we adopt an open-ended information browsing paradigm inspired by realistic human behavior. Annotators begin with a seed factoid claim and use the Google Search API to retrieve a search engine results page (SERP). They then iteratively select relevant web pages, extract new "bridge" claims, and expand the context around the original claim through a retrieval-and-extraction loop. This process is manually performed for 500 samples. To enforce reasoning complexity, we require that each example integrates at least three non-redundant claims or conditions that must be jointly considered to identify the correct answer—ensuring that the final query cannot be resolved by any single piece of information alone. Once a sufficient set of intermediate claims is collected, we use GPT-4o to synthesize a coherent question–answer pair that reflects embedded multi-step reasoning. Human annotators further verify each generated example to ensure there is no information leakage or ambiguity, guaranteeing high-quality supervision signals that reflect realistic, layered information-seeking behavior. An overview of our annotation interface is provided in Figure 3 and Figure 4.

**Scaling via Automated Annotation:** Analysis of the initial 500 manually annotated samples reveals that maintaining consistency and logical coherence within the browsing trajectories is critical for high-quality data. To enable scalability while preserving trajectory quality, we distill effective

Table 1: Main experimental results. The best results are highlighted in **bold**, and second-best results are underlined. For RQRAG and Self-RAG, we use their official 7B checkpoints. All other baselines, including InForage, are built on Qwen-2.5-3B-Instruct. All RAG-based methods apply top-$k = 3$. InForage is trained on the training sets of datasets denoted with $^*$.

| Method | NQ$^*$ | TQA | PopQA | HQA$^*$ | 2Wiki | Mus | Bamb | Self$^*$ | Ave. |
|---|---|---|---|---|---|---|---|---|---|
| **Non-RAG Methods** | | | | | | | | | |
| Vanilla | 12.1 | 28.8 | 13.0 | 15.9 | 24.8 | 2.1 | 2.4 | 4.0 | 12.9 |
| SFT | 24.2 | 26.3 | 12.0 | 20.1 | 25.2 | 6.1 | 13.2 | 8.1 | 16.9 |
| Reasoning | 24.0 | 40.2 | 15.2 | 20.8 | 28.0 | 8.1 | 16.5 | 6.7 | 19.9 |
| **RAG Methods** | | | | | | | | | |
| RAG | 34.8 | 54.4 | 38.7 | 25.5 | 22.6 | 4.7 | 8.0 | 23.4 | 26.5 |
| IRCoT | 11.1 | 31.2 | 20.0 | 16.4 | 17.1 | 6.7 | 24.0 | 0.8 | 15.9 |
| RQRAG | 32.6 | 52.5 | 39.4 | 28.5 | 30.7 | 10.1 | 12.9 | 28.2 | 29.4 |
| Self-RAG | 36.4 | 38.2 | 23.2 | 15.7 | 11.3 | 3.9 | 5.6 | 24.0 | 19.8 |
| Search-o1 | 23.8 | 47.2 | 26.2 | 22.1 | 21.8 | 5.4 | 32.0 | 36.7 | 26.9 |
| Search-R1-PPO | 32.3 | 53.7 | 36.4 | 30.8 | 33.6 | 10.5 | 31.5 | 25.6 | 31.8 |
| Search-R1-GRPO | 40.9 | 55.2 | 40.5 | 34.5 | 36.9 | 15.4 | 32.0 | 29.2 | 35.6 |
| **InForage** (Ours) | **42.1** | **59.7** | **45.2** | **40.9** | **42.8** | **17.2** | **36.0** | **44.1** | **41.0** |

prompting strategies from these manually annotated examples. We then leverage `GPT-4o` as an autonomous agent to automate and scale up the annotation process, systematically generating 20,000 coherent reasoning trajectories, each culminating in a complex question–answer pair. The final dataset comprises a structured split of 19,500 training examples and 500 evaluation examples. The 500 evaluation samples—denoted as the **Self** dataset—comprise real-time, open-ended web tasks that demand multi-hop reasoning, offering a challenging benchmark for search-enhanced reasoning.

**Diverse Web Corpus for Knowledge Freshness:** To mitigate the risk of prior knowledge memorization within the language model, we exclusively crawl web pages published between January 1, 2025, and March 31, 2025. The resulting corpus spans diverse domains—including science, technology, health, culture, and general knowledge—to ensure comprehensive topical coverage. After rigorous filtering for quality and relevance, we retain approximately 80,000 high-quality web pages. We employ `Qwen2.5-72B`, a strong open-source language model, to systematically extract structured factoid claims from each page, resulting in a total of 172,000 validated claims used as seed inputs and intermediate annotations.

**Golden Evidence for Structured Supervision:** Crucially, each annotated example includes explicit records of the *golden web URLs* that serve as the verified factual basis for the final question–answer pairs. This meticulous documentation facilitates precise evaluation of retrieval performance during training. Retrieval actions that successfully identify evidence from these golden URLs are rewarded accordingly, directly incentivizing effective intermediate retrieval behaviors alongside accurate final-answer generation and overall reasoning efficiency.

## 3.4 Main Experiment

In Table 1, we present the main experimental results, from which we have several key findings: (1) InForage consistently outperforms all baselines across both in-domain and out-of-domain datasets, demonstrating strong robustness and generalizability. This includes superior performance not only on standard QA benchmarks like NQ and TriviaQA but also on more challenging settings such our self-constructed web QA set, which require real-time, multi-step reasoning over open-ended knowledge sources. (2) Search-enhanced reasoning methods (e.g., Search-R1 and InForage) show particular strength on multi-hop tasks such as 2Wiki and MuSiQue. Their ability to iteratively decompose queries, formulate subgoals, and retrieve contextually relevant information mirrors human-like problem-solving behavior and leads to consistently stronger results compared to static, one-shot retrieval pipelines. InForage further benefits from reward-guided reasoning, which encourages the generation of effective subqueries and efficient trajectories. (3) While prior retrieval-augmented methods (e.g., Self-RAG, RQRAG) often perform well on narrow tasks with clearly stated queries, their performance degrades under ambiguity or evolving information needs. In contrast, InForage demonstrates greater resilience across task types, confirming our hypothesis that adaptive, reasoning-

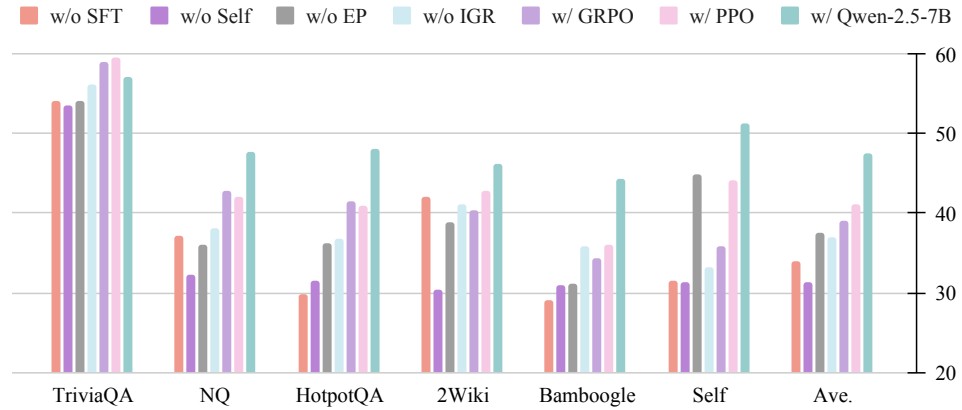

Figure 2: Ablation study of InForage. w/o SFT and w/o Self denote the removal of the supervised fine-tuning stage and the exclusion of the self-constructed dataset, respectively. w/o EP and w/o IGR indicate the ablation of the efficiency penalty and information gain reward. w/ GRPO and w/ PPO compare the use of two different reinforcement learning algorithms. w/ Qwen-2.5-7B refers to replacing the foundation model with Qwen-2.5-7B.

Table 2: Case study using InForage. For clarity, we present only the key content extracted from each retrieved document within the <info> blocks.

**Query**: Which tennis player with more than ten major titles has excelled at a recent top Florida tournament and achieved a significant ranking milestone?
**Ground-truth target**: Novak Djokovic     **RAG**: Sofia Kenin     **Gemini 2.5 Pro (search)**: Novak Djokovic

`<think>` First, identify players who have won more than ten major (Grand Slam) titles. `</think>`
`<search>` Which tennis players have more than ten Grand Slam titles? `</search>`
`<info>` Novak Djokovic, Rafael Nadal, and ... have each won more than ten Grand Slam titles. `</info>`
`<think>` Next, determine a top-tier tennis tournament held in Florida that could serve as a basis for evaluating recent performance. `</think>`
`<search>` What is a major tennis tournament held in Florida? `</search>`
`<info>` The Miami Open is a high-profile annual tennis tournament held in Florida. `</info>`
`<think>` Check which of the shortlisted players (Djokovic, Nadal ...) has recently performed well at the Miami Open. `</think>`
`<search>` Who among Djokovic, Nadal... has recently recorded wins at the Miami Open? `</search>`
`<info>` Novak Djokovic has recently won matches at the Miami Open. `</info>`
`<think>` Finally, confirm whether this player has achieved a notable long-term ranking milestone that sets them apart in ATP history. `</think>`
`<search>` Has Novak Djokovic reached any major career ranking longevity milestones? `</search>`
`<info>` Novak Djokovic has surpassed 1000 weeks ranked within the top 100. `</info>`
`<answer>` Novak Djokovic `</answer>` ←**InForage's Answer.**

aware retrieval is crucial for complex tasks. Its principled integration of retrieval into the reasoning loop, guided by outcome, information gain, and efficiency signals, enables the model to dynamically navigate complex knowledge spaces more effectively than static or manually designed strategies.

### 3.5 Discussion

**Ablation Study**    To evaluate the contributions of InForage's key design components, we conduct comprehensive ablation studies, as summarized in Figure 2. Our findings are as follows: (1) Training Settings: Leveraging our self-constructed dataset for both supervised fine-tuning (SFT) and reinforcement learning (RL) yields consistent performance improvements. This confirms that high-quality, reasoning-aligned training data is essential for search-enhanced reasoning methods—particularly, SFT provides a strong initialization that benefits subsequent RL optimization. (2) Reward Design: Ablating the information gain reward (IGR) leads to a consistent performance drop across datasets, validating its effectiveness in promoting meaningful intermediate retrievals. Removing the efficiency penalty (EP) also degrades performance, except on our self-constructed dataset. This suggests that

complex, real-world information-seeking tasks may inherently require longer reasoning chains, making the balance between step efficiency and answer completeness task-dependent. (3) RL Algorithms: Comparing PPO and GRPO, we observe that PPO generally yields better results. While GRPO occasionally outperforms PPO on specific datasets, PPO's use of learned reward signals (rather than rule-based estimates) appears better suited for evaluating nuanced reasoning quality in complex tasks. (4) Model Scaling: Replacing the 3B foundation model with Qwen2.5-7B leads to further gains across most benchmarks, demonstrating that InForage effectively scales with model capacity and can harness larger LLMs for improved performance.

**Case Study**    To concretely illustrate the reasoning process of InForage, we conduct a case study on a test example from our self-constructed dataset. As shown in Table 2, the input query asks about a tennis player who satisfies multiple overlapping conditions, each corresponding to several potential candidates. Only by combining all the constraints can the correct answer be uniquely identified. This type of embedded information-seeking task poses a significant challenge for traditional RAG methods, which rely on one-pass retrieval and often miss critical evidence. As a result, vanilla RAG fails to find the correct answer. In contrast, InForage accurately decomposes the layered intent of the query, iteratively retrieves relevant evidence, and successfully identifies the correct answer—demonstrating its strong search-enhanced reasoning capability. For comparison, Gemini 2.5 Pro, a state-of-the-art LLM with built-in search tools, also produces the correct answer. However, InForage achieves this using a much smaller 3B model, highlighting its efficiency and effectiveness.

## 4    Related Work

**Retrieval-Augmented LLMs:**    Despite the rapid progress of LLMs [OpenAI, 2023, Group, 2025], their inherent limitations—such as hallucination and outdated knowledge—remain obstacles to real-world deployment[Gao et al., 2024, Zhao et al., 2024a]. Retrieval-Augmented Generation (RAG), first proposed by Lewis et al. [2020b], addresses these challenges by equipping LLMs with external retrieval capabilities to provide contextually relevant, up-to-date information [Izacard and Grave, 2021, Gao et al., 2024]. This paradigm not only improves factual accuracy but also enhances temporal relevance. Subsequent research has focused on two fronts: improving retrieval quality to raise the generation ceiling [Qian et al., 2024a, Gao et al., 2024], and optimizing the integration of retrieved content into the generative process [Jiang et al., 2023, Zhao et al., 2024b]. However, traditional RAG methods typically assume well-defined queries and structured knowledge, limiting their scope to simple factoid QA [Nogueira and Cho, 2020, Lewis et al., 2020b]. Recent work has emphasized the inadequacy of this static setup for real-world tasks, where information needs are often implicit, ambiguous, or evolving [Qian et al., 2024b]. These settings demand adaptive, multi-step retrieval guided by iterative reasoning. To this end, retrieval-augmented reasoning has emerged, with recent innovations embracing more dynamic and structured approaches: GraphRAG [Edge et al., 2024] and HippoRAG [Jimenez Gutierrez et al., 2024] enhance global awareness through graph-based representations, while agent-driven systems like ActiveRAG [Xu et al., 2024] integrate planning and evidence aggregation. Together, these advances highlight a growing need for RAG systems that move beyond static retrieval—toward reasoning-aware, goal-driven interaction with external knowledge to meet the demands of complex, open-ended tasks.

**Reasoning LLMs:**    Enhancing the reasoning capabilities of LLMs has become a central focus in advancing their ability to tackle complex, real-world tasks. Progress has been made across the entire training pipeline: pretraining on code and mathematical data fosters structured thinking [Wei et al., 2022]; supervised fine-tuning on reasoning-rich datasets improves response fidelity; and alignment through reinforcement learning and preference modeling refines multistep reasoning behavior [Gulcehre et al., 2023, Kumar et al., 2024, Zhang et al., 2024]. In parallel, prompting strategies—such as chain-of-thought [Wei et al., 2022], tree-of-thought [Yao et al., 2024], and ReAct [Yao et al., 2023]—guide models to explicitly decompose problems and explore multiple solution paths. Notably, reasoning-centric models like OpenAI's o1 and DeepSeek R1 have achieved impressive performance on challenging domains such as mathematical problem-solving and coding, where iterative reflection and progressive refinement are key [OpenAI, 2024, DeepSeek-AI, 2025].

However, these models typically operate on static internal knowledge, limiting their adaptability in knowledge-sparse or dynamic contexts. To overcome this constraint, recent research has shifted toward search-augmented reasoning, which equips LLMs with the ability to issue subqueries and

iteratively gather external evidence during inference. For instance, Search-R1 [Jin et al., 2025a] introduces a reinforcement learning framework that teaches LLMs when and how to interact with search engines. Search-o1 [Li et al., 2025] integrates retrieval directly into the o1 reasoning loop and introduces a Reason-in-Documents module to filter noisy content before reintegration. Extending further into open-world settings, DeepResearcher [Zheng et al., 2025] applies end-to-end RL to train agents that can navigate unstructured web environments, demonstrating emergent behaviors such as planning, evidence synthesis, and self-correction.

Building on this evolving landscape, we propose InForage, a framework that models search-enhanced reasoning as an information foraging process. Unlike prior methods that reward only final answers, InForage optimizes the full reasoning trajectory—encouraging effective subqueries, relevant retrievals, and efficient solutions. By bridging cognitive theory and reinforcement learning, it offers a more adaptive and holistic approach to retrieval-augmented reasoning.

## 5 Conclusion

This paper presents InForage, a reinforcement learning framework that advances search-augmented reasoning in LLMs by drawing on principles from Information Foraging Theory. Rather than treating retrieval as a static, pre-inference step, InForage integrates it dynamically into the reasoning process, rewarding not only correct final answers but also informative intermediate retrievals and concise solution paths. This structured optimization encourages models to emulate human-like information-seeking behaviors by formulating subqueries, evaluating partial evidence, and refining reasoning trajectories. Empirical results across a broad range of benchmarks show that InForage consistently surpasses all baselines. Its iterative reasoning capabilities prove especially effective on complex queries requiring layered evidence integration, demonstrating robust generalization across task types. Moreover, InForage exhibits strong scalability with larger model sizes and comprehensive analyses validate the effectiveness of InForage's method design. Together, these findings highlight the importance of coupling reasoning with retrieval in a learning-driven, context-sensitive manner. InForage offers a unified and principled solution that closes the gap between static retrieval pipelines and the dynamic, adaptive reasoning needed for real-world knowledge-intensive applications.

## Acknowledgement

This work was supported by National Natural Science Foundation of China No. 62502049.

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

# A  Prompts

Below we present the core prompts used in this paper. The first prompt is designed for search-enhanced reasoning, guiding the model to perform step-by-step retrieval and inference. The second prompt is used to generate question–answer pairs from a batch of extracted factual claims. For completeness, additional prompts—such as those used for claim extraction—are provided in the source code included in the supplementary materials.

---

**Prompt for Search-Enhanced Reasoning**

You are an intelligent agent designed to solve complex queries or tasks by retrieving external knowledge and reasoning step by step.
Please follow these instructions carefully:
1. For each piece of information you receive:
- Think step by step and explain your reasoning inside <think> and </think> tags.
2. If you need more information to proceed:
- Issue a search by writing your subquery inside <search> and </search> tags.
- Retrieved results will appear between <evidence> and </evidence> tags.
- You can conduct multiple searches as needed.
3. When you have collected enough information:
- Provide your final answer using the <answer> and </answer> tags.
- Do not include explanations or reasoning in the answer block.
- Keep your answer concise.
Now, solve the following task:
Task: {question}

---

**Prompt for Query-Answer Pair Generation**

Based on the following evidence, generate a complex multi-hop query that requires connecting multiple pieces of information to answer.
Create a challenging question that requires reasoning across multiple facts. The question should be specific enough that it can only be answered by connecting several pieces of evidence together.
For example, given the evidence list: {Example Evidence List}
The multi-hop query could be: {Example Query}
Requirement:
1. Ensure Coherence: Make sure the question flows logically from the combined information and is clear and unambiguous
2. Formulate the Question: Create a question that cannot be answered by relying on just one of the sentences but instead requires understanding and linking the information from all of the sources.
3. Ensure Multi-hop: The question should require at least 3 logical steps to answer.
4. The answer should be specific and short.
Now, based on the following evidence, generate a multi-hop query that requires at least 3 logical steps to answer:
{evidence_str}
Generate a multi-hop query that requires at least 3 logical steps to answer. Output in the following format: {"query": "multi-hop query", "answer": "answer"}

---

# B  Limitations and Broader Impact

**Limitations.**   In this work, we introduce InForage, a reinforcement learning framework that enhances search-augmented reasoning by aligning retrieval actions with evolving reasoning goals, inspired by Information Foraging Theory. While our results demonstrate strong performance across a range of QA tasks, several limitations remain.

First, due to computational constraints, our experiments primarily focus on Qwen2.5-3B and 7B models. We do not explore larger foundation models or alternative model families (e.g., LLaMA, Mixtral), which could further enhance performance. Future work will expand evaluation across more model scales and architectures to validate generalizability.

Second, while InForage shares high-level goals with emerging search-enhanced reasoning models, many of these works are still in progress or unpublished at the time of our submission. As such, we do not include all of them in our experimental comparisons, though we provide design-level discussions to highlight conceptual differences.

Third, our self-constructed dataset focuses on QA tasks with short-form answers to ensure verifiability—crucial for rule-based reward assignment. While we believe the InForage framework is extensible to complex tasks such as long-form synthesis, report writing, and multi-document summarization, these applications are not explored in this paper.

**Broader Impact.** The central contribution of this paper is the shift of retrieval from a static, pre-inference step to a dynamic, reasoning-integrated process. This mirrors how humans seek information—thinking, searching, and composing in parallel—and offers a more flexible and cognitively aligned paradigm for knowledge-intensive tasks. We believe this capability can generalize to domains where adaptive reasoning is essential, including scientific writing, investigative journalism, legal analysis, and knowledge curation.

Moreover, while we focus on retrieval as the primary external tool, the proposed framework and dataset pipeline are tool-agnostic. Our methodology can be extended to optimize interactions with other tools such as code compilers, search APIs, or structured databases. By offering a complete pipeline—from dataset construction to multi-stage reward design and reinforcement learning—InForage lays the groundwork for building general-purpose tool-augmented reasoning agents that are better aligned with real-world human workflows.

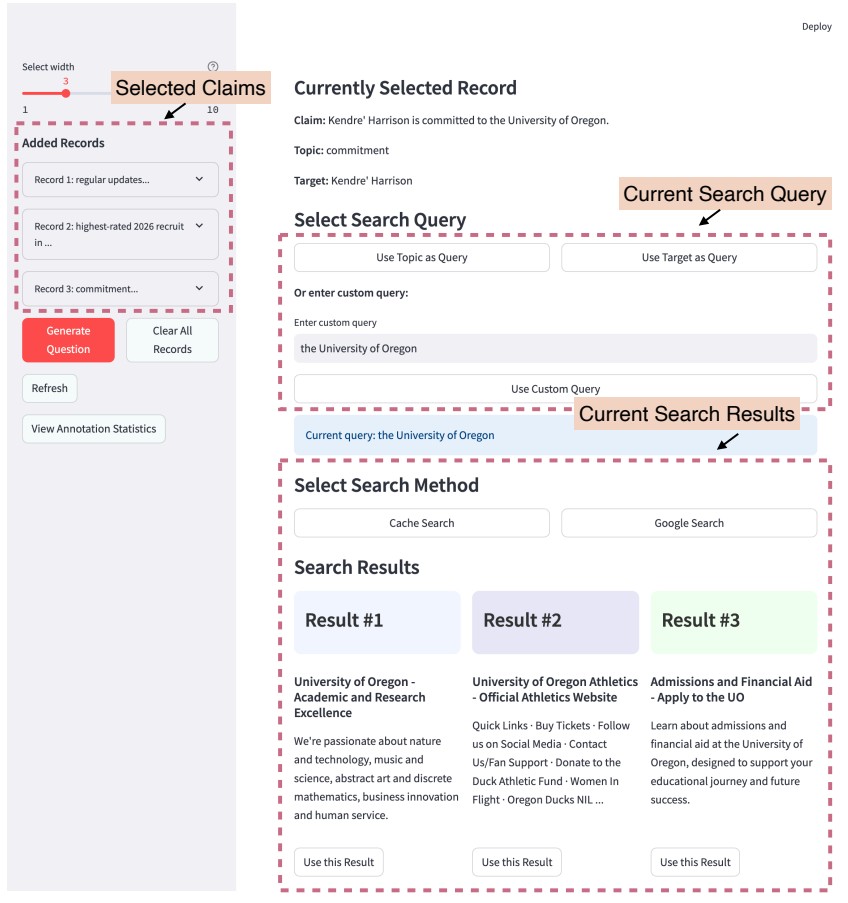

Figure 3: Annotation page for construct the training dataset.

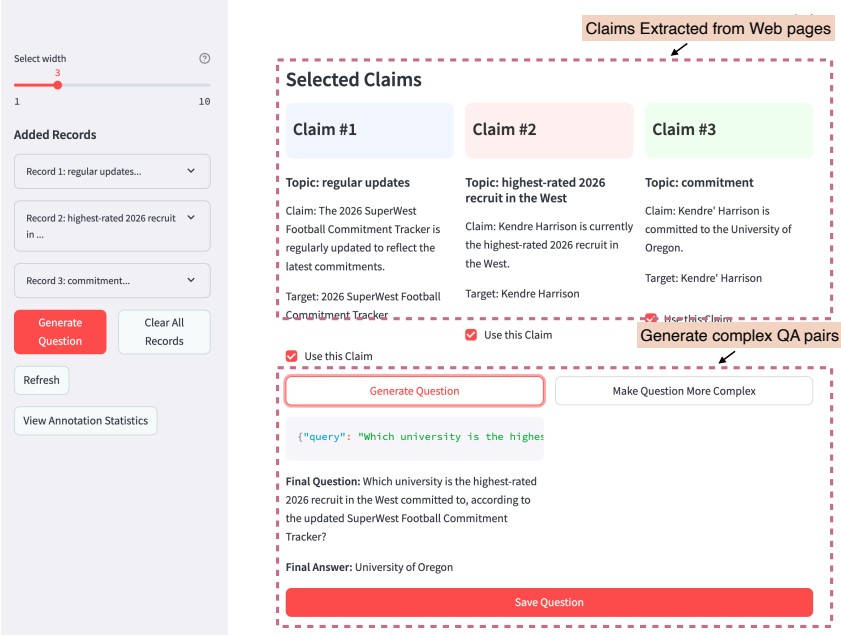

Figure 4: Annotation page for construct the training dataset.

