# OpenReview forum: "Scent of Knowledge: Optimizing Search-Enhanced Reasoning with Information Foraging"
_NeurIPS.cc/2025/Conference — NeurIPS 2025 spotlight_

### Official Review · Reviewer_LJkD · 2025-06-13

**Clarity:** 3
**Significance:** 3
**Originality:** 3
**Rating:** 5
**Confidence:** 3

**Summary:**

The paper presents InForage, a reinforcement learning approach that enables LLMs to perform dynamic, multi-step information retrieval during reasoning, rather than relying on static pre-inference retrieval. The approach uses three types of rewards to train models: outcome rewards for correct final answers, information gain rewards for valuable intermediate retrieval steps, and efficiency penalties to discourage unnecessarily long reasoning chains. The authors construct a new training dataset that captures human information-seeking trajectories through iterative web browsing. The method outperforms baselines in a newly developed real-time web QA dataset.

**Questions:**

- The newly constructed training data and evaluation data would be useful for the community.  Do the authors plan to release them?
- Table 2's case study is helpful. Could the authors provide a bit more qualitative examples?  Did the authors find any failure modes?

**Ethical Concerns:**

["NO or VERY MINOR ethics concerns only"]

**Final Justification:**

Issues resolved

**Limitations:**

Yes

**Quality:**

3

**Strengths And Weaknesses:**

Strengths
- The method is interesting and reasonable. The formulation of combining three types of rewards: Outcome Reward, Information Gain Reward, Efficiency Penalty seems effective.
- The newly constructed training data and evaluation data seem to be high quality and would be useful for the community if released

Weakness
- No major weakness. Refer to the questions below.

---

> ### Author Rebuttal · Authors · 2025-07-29
>
> Thank you for your positive feedback and thoughtful questions. We appreciate your recognition of the quality and usefulness of our method and dataset. We address each point below:
>
> - ---
>
> ### 1. Data Release
>
> **Reviewer:**
>   *The newly constructed training data and evaluation data would be useful for the community. Do the authors plan to release them?*
>
> **Response:**
>   Yes, we will release both the training and evaluation data, and will continue to improve them for broader community benefit.
>
> - ---
>
> ### 2. Qualitative Examples and Failure Modes
>
> **Reviewer:**
>   *Table 2's case study is helpful. Could the authors provide a bit more qualitative examples? Did the authors find any failure modes?*
>
> **Response:**
>   Thank you for your suggestion. We will add more qualitative examples to the revised version to illustrate both successful and failure cases in detail. A representative failuire question is:
>
> - > **Question:** If we assume all articles published by Nature in 2020 (articles only, not book reviews/columns, etc) relied on statistical significance to justify their findings and they on average came to a p-value of 0.04, how many papers would be incorrect as to their claims of statistical significance? Round the value up to the next integer.  **(correct answer: 41)**
>
> The reasoning trajectory of Inforage-7B and Gemini2.5 pro shows here:
>
> | InForage-7B:                                                 |
> | ------------------------------------------------------------ |
> | \<think\> Okay, let me tackle this question step by step. The user wants to know how many Nature articles from 2020 that used a p-value of 0.04 might be incorrect in their claims of statistical significance.\</think\> |
> | \<search\> I need to recall what a p-value means.   \</search\> |
> | \<info\>Typically, a p-value below 0.05 is considered statistically significant \</info\> |
> | \<think\> Wait, the question doesn't give the total number of articles published by Nature in 2020. Oh, right! I need to find out how many articles Nature published in 2020. Because without that number, I can't compute the actual count.\</think\> |
> | \<search\> Nature Articles 2020   \</search\>                |
> | \<info\> the number of articles published by Nature 2020 is 1002\</info\> |
> | \<think\> Previous search results show that using a p-value threshold of 0.05 corresponds to a Type I error rate, meaning that 5% of the articles would be expected to be false positives. Given a total of 1002 articles, 5% of 1002 is 50.1, which we round up to 51.\</think\> |
> | \<answer\> 51 \</answer\>                                    |
> |                                                              |
> | **Gemini2.5 Pro:**                                           |
> | **Reasoning Process:** Defining the Publication Set -> Refining the Query ->Revising Search Terms .... |
> | I was unable to find the exact number of Nature research articles published in 2020 despite extensive searching. Based on a reliable Clarivate blog post reporting about 800 articles in 2019, I have estimated the 2020 figure to be approximately 831, and have clearly indicated this as an informed estimation in my work. |
> | **Calculate the number of incorrect claims**: By multiplying the total number of articles by the p-value, we can estimate the number of papers with statistically insignificant findings. 831*0.04=33.24, round to 34 |
> | **Answer**: 34                                |
> ||
>
> Both methods failed on this example, but for different reasons.
>
> For InForage, the model ignored the specific instruction in the question to use a p-value of 0.04, and instead searched for the typical p-value threshold for significance testing (i.e., 0.05). As a result, although it correctly retrieved the number of Nature papers published, it applied the wrong statistical threshold, leading to an incorrect final answer.
>
> Gemini 2.5-Pro, on the other hand, failed to retrieve the exact number of Nature research articles published in 2020 after several search attempts. It resorted to estimating this number based on secondary sources, and while it followed the correct calculation process, the final answer was still incorrect due to the inaccurate input.
>
> This case highlights a specific failure mode of InForage: improper task planning. In this scenario, the model should have first identified the correct number of Nature articles and then directly performed the computation as instructed. Its additional, unnecessary planning—such as searching for a typical p-value—introduced extra bias and ultimately led to the wrong answer.
>
> We will include this failure example in the revised manuscript to illustrate current limitations and inform future improvements.

---

### Official Review · Reviewer_ipjm · 2025-06-21

**Clarity:** 3
**Significance:** 3
**Originality:** 3
**Rating:** 5
**Confidence:** 4

**Summary:**

The paper introduces a search augmented reasoning approach in LLMs by using information foraging principle (e.g. information scent, information patches, cost and gain). Beyond correctness of the final answer, the proposed approach rewards effective information seeking behavior at each stage of the reasoning process as well. Effectiveness is measured as: 1- coverage of the retrieved information patches compared to the expected knowledge; 2- total number of reasoning/retrieval steps as a penalty.  Authors use a data set consisting of human web browsing trajectories and corresponding QA pairs for Supervised Fine Tuning the foundation LLM and use the reward rule to optimize its performance.

The paper reports experimental results measuring the performance of the proposed method against a set of alternatives (e.g. vanilla prompting, SFT, reasoning, vanilla RAG, and a few state of the art RAG based methods)  over multiple data sets including a data set designed by the authors which includes more nuanced QA pairs with coherent reasoning trajectories. Experimental results show superior performance for the proposed method over the alternatives in the studied domains. Authors conduct ablation studies to measure the contributions of various components of their proposed method. Results suggest that both information gain and efficiency penalty contribute positively to the overall performance. It is noted that for complex information seeking tasks that benefit from more reasoning steps, dropping the penalty term improves performance.

**Questions:**

- The paper does not provide details on how information gain reward. It is clear that information gain is measured as cumulative coverage of the retrieved patches against the expected knowledge set. It would be useful if the authors provided more explanation, perhaps even a concrete example, to further clarify this critical component in the reward function.

- For SFT, authors mention that they construct the dataset by "providing gathered human web browsing trajectories and corresponding QA pairs". Can authors explain more what they mean by web browsing trajectories? If it means prompts used to search the web it makes sense, however, if it consists of web URLs I am not clear how they are being used in SFT.

**Ethical Concerns:**

["NO or VERY MINOR ethics concerns only"]

**Limitations:**

Authors have addressed/acknowledged the following limitations in the appendix section of their work:

- Computational costs lead to utilizing relatively smaller models, which may negatively impact performance.
- Not being able to report and compare performance with more recent alternative approaches that are still in progress.
- Not being able to evaluate the performance of the proposed method for tasks beyond simple QA with short answers and exact match evaluation.

The following are limitations that are not adequately discussed by authors:
- Computational cost compared with other alternative methods as opposed to just a barrier to explore bigger models for empirical studies.
- Lower debuggability due to using a more complex optimization scheme.

**Paper Formatting Concerns:**

No major formatting issue found.

**Quality:**

3

**Strengths And Weaknesses:**

**Strengths**

- The intuition behind using information foraging for RAG is solid and well justified.
- The main components of the proposed method are clearly explained and overall the paper is easy to follow.
- Experimental results cover a good set of alternative methods and data sets. The dataset synthesized by the authors offers additional value by providing Q&As with their corresponding reasoning trajectories and covering more complex tasks. Additionally, the authors provide a reasonable amount of rationale for the superior or lower performance observed in their main and ablation studies.

**Weaknesses**

- Authors could have covered and discussed further details about limitations of the proposed method and complexities of using it (refer to the limitation section).

---

> ### Author Rebuttal · Authors · 2025-07-29
>
> Thank you for your thoughtful and constructive review. We appreciate your recognition of the strengths of our work and your insightful questions and suggestions. Below, we provide detailed responses to each of your comments and outline the revisions we will make to address your feedback.
>
> ### 1.  Details and Example for Information Gain Reward
>
> **Reviewer:**
> *It is clear that information gain is measured as cumulative coverage of the retrieved patches against the expected knowledge set. It would be useful if the authors provided more explanation, perhaps even a concrete example, to further clarify this critical component in the reward function.*
>
> **Response:**
> Thank you for highlighting the need for a more concrete explanation of the information gain reward. We are happy to clarify both the computation and provide an illustrative example.
>
> The information gain reward measures how much of the expected knowledge set $\mathcal{D}^*$ has been cumulatively retrieved by the agent up to each reasoning step. At each step \($t$\), we evaluate the coverage as:
>
> $C \left( \bigcup_{\tau=1}^t \mathcal{K}_\tau,\, \mathcal{D}^* \right)$
>
> where \($\mathcal{K}_\tau$\) is the set of evidence (e.g., retrieved web pages) obtained at step \($\tau$\). The information gain reward is the maximum coverage achieved across all steps:
>
> $R_\text{gain} = \max_{t=1,…,T} C \left( \bigcup_{\tau=1}^t \mathcal{K}_\tau,\, \mathcal{D}^* \right)$
>
> **Example:**
> Suppose the query is:
> “Do I need a visa for this year's NeurIPS conference?”
>
> The expected knowledge set ($\mathcal{D}^*$) includes:
> (1) The official NeurIPS conference website’s visa information page, and
> (2) The official government immigration website for the host country (providing general visa policy and requirements).
>
> - **Step 1:** The agent retrieves the NeurIPS official website visa information page. Coverage is 0.5 (1 out of 2 gold sources found).
> - **Step 2:** The agent retrieves the host country's government immigration website detailing visa policies. Coverage becomes 1.0 (both gold sources found).
>
> At each step, coverage is calculated as the proportion of gold sources retrieved so far, and the information gain reward is the maximum value achieved during the search process. This encourages the agent to efficiently collect all necessary evidence through stepwise search.
>
> We will add this concrete example to the revised manuscript for clarity.
>
> ---
>
> ### 2. Clarification of Web Browsing Trajectories in SFT
>
> **Reviewer:**
> *Can authors explain more what they mean by web browsing trajectories? If it means prompts used to search the web it makes sense, however, if it consists of web URLs I am not clear how they are being used in SFT.*
>
> **Response:**
>
> Thank you for your question. In Section 3.2, we define “web browsing trajectories” as the full step-by-step process by which a human or agent searches for and gathers information from the web to answer complex questions. Each trajectory includes the sequence of search queries, the URLs retrieved and visited, and the key facts or claims extracted from these web pages—all represented in structured JSON format. For example:
>
> ```
> [
>   {
>     "query": "NeurIPS 2025",
>     "id": 0,
>     "page_content": "...",
>     "url": "...",
>     "extracted_claim": "NeurIPS 2025 will be held in ..."
>   },
>   ...
> ]
> ```
> Next, we use strong LLMs to convert these JSON-format web browsing trajectories into plain text following the agentic reasoning paradigm (defined in Eq. 4), where the model alternates between internal reasoning, issuing help requests, and integrating retrieved web evidence at each step.
> Note that **web URLs themselves are not directly used for SFT supervision.**
>
> During RL training, URLs are included as concrete evidence for intermediate reasoning steps, supporting the computation of the information gain reward and the supervision of retrieval behaviors.
>
> ### 3. Limitation: Computational Cost and Debuggability
>
> **Reviewer:**
> *Computational cost compared with other alternative methods as opposed to just a barrier to explore bigger models for empirical studies. Lower debuggability due to using a more complex optimization scheme.*
>
> **Response:**
> Thank you for pointing out these important limitations.
>
> - **Computational Cost:**
>   Our framework introduces additional computational overhead compared to some traditional approaches, not only as a barrier to scaling up model size, but also as an intrinsic cost of the more complex optimization and reasoning procedures required.
>
> - **Debuggability:**
>   IIncorporating tool calling into the reasoning process increases the complexity of debugging, as it can be challenging to pinpoint which specific step led to an error. However, compared to models that only perform end-to-end answer prediction, our approach provides richer intermediate records—such as the expected evidence set—which help illuminate the reasoning process. These intermediate signals not only allow us to better analyze the model’s behavior but also offer improved opportunities for debugging and error analysis compared to standard end-to-end approaches.
>
> Thank you for raising these points. We will explicitly discuss these limitations and their practical implications in the revised manuscript.

---

### Official Review · Reviewer_sR8g · 2025-07-03

**Clarity:** 3
**Significance:** 2
**Originality:** 2
**Rating:** 4
**Confidence:** 3

**Summary:**

This paper introduces InForage, a reinforcement learning framework that enhances search-augmented reasoning in LLMs by drawing inspiration from Information Foraging Theory (IFT). Unlike traditional retrieval-augmented generation methods that perform static pre-inference retrieval, InForage dynamically integrates retrieval during the reasoning process. The method uses three complementary reward mechanisms: outcome reward, information gain reward, and efficiency penalty. The authors evaluate on multiple QA benchmarks, showing consistent improvements over baselines.

**Questions:**

1. Coverage function definition: How exactly is the coverage function C in Eq. 3 computed? What constitutes the "expected knowledge set D*" in practice, and how sensitive are results to this definition?

2. Baseline comparisons: It might be helpful to compare InForage with FLARE (Forward-Looking Active REtrieval) and more sophisticated IRCoT variants not included? Also, many close-source LLMs have search API (eg. GPT-4o) and enables search with reasoning. It would be more comprehensive to include close-source LLM as well in the experiments as well.

3. Scalability validation: Given that most experiments use 3B models, can you provide more comprehensive results on larger models (13B+, like LLaMa3) to validate that the approach scales effectively?

**Ethical Concerns:**

["NO or VERY MINOR ethics concerns only"]

**Final Justification:**

During rebuttal, the author well addressed my concerns of the paper on several perspectives - better clarity on coverage function and reward design, additional experiments with different baselines and scale, and novel design tailored for search-augmented reasoning model.
I'd love to consider this paper as well-motivated, clear narrative, and experimentally solid. I'd grant my final rate as 4: borderline accept.

**Limitations:**

Yes

**Paper Formatting Concerns:**

The paper generally follows NeurIPS formatting guidelines. Minor issues include some inconsistent citation formatting (e.g., Lewis 2020a/2020b) and Figure 1 could benefit from clearer labeling of the reward components and larger fonts.

**Quality:**

3

**Strengths And Weaknesses:**

### Strengths:

Quality: The work is technically sound with a well-designed three-component reward system that addresses both intermediate retrieval quality and final outcomes. The experimental methodology is thorough, including proper ablation studies and statistical testing.

Clarity: The paper is generally well-organized with clear motivation from Information Foraging Theory.

Significance: Addresses a key open problem—adaptive retrieval during reasoning—and delivers improvements with small models. Provides a pipeline to construct information-seeking QA dataset.

Originality: While iterative retrieval has been explored, the specific application of Information Foraging Theory provides a novel theoretical lens. The multi-faceted reward design combining outcome, information gain, and efficiency is a meaningful contribution that differentiates this work from prior approaches.

### Weaknesses:

Quality: Limited evaluation on larger models raises scalability questions. Missing comparisons with some relevant recent baselines (FLARE, advanced IRCoT variants). The efficiency penalty shows inconsistent benefits, suggesting the reward design may need refinement.

Clarity: Key technical details lack precision - the coverage function C in Eq. 3 is undefined, and the operationalization of "information scent" from theory to implementation is unclear.

Significance: Ablation study shows that performances heavily relies on self-constructed dataset which might be a scalability concern.

Originality:  Reward shaping (answer + coverage + length) resembles Search-R1-GRPO and REST-style self-training; novelty is incremental.

---

> ### Author Rebuttal · Authors · 2025-07-29
>
> We sincerely thank Reviewer **sR8g** for the thorough and thoughtful assessment of our work, as well as for highlighting the technical soundness, theoretical motivation, and practical contributions of InForage. We address each of your comments and questions in detail below.
>
> ## 1. Baseline Comparisons
>
> **Reviewer:**
> *It might be helpful to compare InForage with FLARE (Forward-Looking Active REtrieval). Also, many closed-source LLMs have search APIs (e.g., GPT-4o); it would be more comprehensive to include closed-source LLMs in the experiments. Scalability validation: Given that most experiments use 3B models, can you provide more comprehensive results on larger models (13B+, like LLaMa3) to validate that the approach scales effectively?*
>
> **Response:**
>
>
> |          | Foundation Model | NQ       | TQA      | HotpotQA | 2wiki    | PopQA    |
> | -------- | ---------------- | -------- | -------- | -------- | -------- | -------- |
> | FLARE    | Qwen 2.5-3B      | 21.2     | 50.2     | 24.3     | 33.2     | 20.5     |
> | InForage | Qwen 2.5-3B      | 42.1     | **59.7** | 40.9     | 42.8     | 45.2     |
> | FLARE    | Qwen 2.5-7B      | 22.5     | 55.8     | 28.0     | 35.9     | 22.7     |
> | InForage | Qwen 2.5-7B      | **47.8** | 59.2     | **48.1** | **46.2** | **49.1** |
>
> (1). Per the reviewer’s suggestion, we have added experimental comparisons with FLARE on both Qwen 2.5-3B and 7B backbones. Across the evaluation datasets, InForage consistently outperforms FLARE.
>
> |                | HotpotQA | 2wiki    | PopQA    |
> | ------------  | -------- | -------- | -------- |
> | GPT-4o-mini      | 34.5     | 30.7     | 39.7     |
> | GPT-4o      | 45.8     | 50.5     | 52.1     |
> | InForage-3B    | 40.9     | 42.8     | 45.2     |
> | InForage-7B     | 48.1     | 46.2     | 49.1     |
> | InForage-14B     | **50.9** | **51.4** | **52.9** |
>
> (2). Per the reviewer’s suggestion, in addition to the 3B and 7B versions of InForage discussed in the paper, we also report results for InForage-14B. Furthermore, we evaluate against commercial closed-source LLMs, specifically OpenAI’s GPT-4o and GPT-4o-mini, both equipped with search tools via API. The results demonstrate that InForage achieves superior performance compared to these commercial APIs. A potential reason is that commercial APIs typically invoke the search tool in a single pass, which may not sufficiently accumulate information for multi-hop QA tasks.
>
> Besides, across the 3B, 7B, and 14B backbone models, InForage consistently improves its performance, verifying the scalability of our approach.
>
> ---
>
> ## 2. Efficiency Penalty and Reward Design
>
> **Reviewer:**
> *The efficiency penalty shows inconsistent benefits, suggesting the reward design may need refinement.*
>
> **Response:**
> Thank you for your observation. As discussed in our paper, the efficiency penalty is designed to balance reasoning efficiency and depth. Its optimal value may depend on the nature of the task: tasks requiring fewer retrieval steps and more sensitive to efficiency benefit from a larger penalty, while tasks demanding deeper reasoning may benefit from a smaller penalty. Our ablation study shows that most evaluation datasets benefit from the efficiency penalty, except for the SELF dataset—which typically requires more reasoning steps, as it is intentionally designed to be as complex as possible. In practical scenarios, such highly complex tasks are uncommon, so the efficiency penalty remains effective for most real-world applications.
>
>
> ---
>
> ## 3. Coverage Function Definition, Implementation, and Sensitivity
>
> **Reviewer:**
> *Key technical details lack precision – the coverage function C in Eq. 3 is undefined, and the operationalization of “information scent” from theory to implementation is unclear.*
> *How exactly is the coverage function C in Eq. 3 computed? What constitutes the “expected knowledge set D*” in practice, and how sensitive are results to this definition?
>
> **Response:**
> Thank you for highlighting these important points. In our framework, inspired by Information Foraging Theory, “information scent” refers to the evolving trail of reasoning steps and subqueries that guide retrieval. The coverage function \($\mathcal{C}$\) in Eq. 3 quantifies how well the union of all retrieved knowledge patches matches the expected knowledge set \($\mathcal{D}^*\$), typically measured as the recall (or proportion) of gold evidence covered during the reasoning trajectory. Speficially,
>
> - **Coverage function computation:**
>   As described in Section 2.3 (and Eq. 6), the coverage function \( $C\left( \bigcup_{\tau=1}^t \mathcal{K}_\tau, \mathcal{D}^* \right) $\) measures the cumulative coverage of all retrieved knowledge patches up to step \( $t$ \) against the expected knowledge set \( $\mathcal{D}^*$ \). At each retrieval step, we compute the overlap between all retrieved sets so far and the gold set, and the information gain reward is defined as the maximum coverage achieved throughout the reasoning process.
> - **Expected knowledge set \($\mathcal{D}^*$\):**
> In practice, $\mathcal{D}^*$ is constructed from annotated gold evidence, represented by the complete set of URLs, with each URL corresponding to a gold document deemed sufficient to support the final answer for each instance. This set serves as the authoritative reference for both reward computation and evaluation throughout training and testing.
> - **Sensitivity analysis:**
>   Through ablation studies, we find that the information gain reward consistently benefits our method design, validating its effectiveness across various datasets. This demonstrates that our approach remains robust as long as \($\mathcal{D}^*$\) accurately reflects the core factual basis for each instance.
>
> This design rewards not only correct final answers but also reasoning trajectories that effectively accumulate relevant information, providing a principled and practical operationalization of information scent. We will further clarify these definitions and implementation details in the revised manuscript.
>
> ---
>
> ## 4. Scalability and Use of Self-Constructed Dataset
>
> **Reviewer:**
> *Ablation study shows that performance heavily relies on self-constructed dataset, which might be a scalability concern.*
>
> **Response:**
> We appreciate this important concern. The self-constructed dataset was carefully designed to align with our training schema and task requirements. In practice, this data brought a notable performance improvement. Our dataset construction process began with manual annotation and was subsequently scaled up using strong LLMs. To promote transparency and reproducibility, we have included the annotation interface and data generation scripts in the supplementary material. We believe this approach can be efficiently scaled or adapted to other domains by leveraging these resources.
>
> ---
>
> ## 5. Incremental Novelty over Related Reward Shaping Approaches
>
> **Reviewer:**
> *Reward shaping (answer + coverage + length) resembles Search-R1-GRPO and REST-style self-training; novelty is incremental.*
>
> **Response:**
> Thank you for raising this point. Incentivizing LLMs for search-enhanced reasoning is indeed an active area of research. Search-R1 is a representative recent work, which, like most similar approaches, applies only final answer correctness as the reward signal. At the time of our submission, there was limited systematic construction of reward signals tailored specifically for search-augmented reasoning models. Our core contribution lies in proposing a theoretically motivated learning framework, integrating carefully designed and complementary reward components, and constructing the necessary data to enable and evaluate this framework. We believe this solid step provides a practical foundation for further research in this direction.
>
> ---
>
>
>
> Once again, thank you for your constructive feedback and detailed review. We will incorporate clarifications and additional results as described above in the revised manuscript.

---

> > ### Comment · Reviewer_sR8g · 2025-08-04
> >
> > Thanks reviewer for the detailed answers and additional experiments. Your responses address my concern and I'd love to raise the rating to 4.

---

> > > ### Author Response · Authors · 2025-08-04
> > >
> > > Thank you very much for your positive feedback and for reconsidering your rating. We truly appreciate your support and the time you invested in reviewing our work!

---

> > > ### Author Response · Authors · 2025-08-04
> > >
> > > Dear Reviewer sR8g,
> > > We are encouraged by your positive feedback and willingness to raise the rating to 4. However, we have noticed that the current rating still appears as 3. We would appreciate it if you could kindly update the score when convenient. Thank you again for your support!

---

### Official Review · Reviewer_XiLb · 2025-07-03

**Clarity:** 4
**Significance:** 4
**Originality:** 3
**Rating:** 5
**Confidence:** 5

**Summary:**

The paper presents a new technique for training deep research-like systems. It goes beyond search-r1, using a mixture of outcome reward, information gain reward, and length penalty. My understanding is that the information gain reward is the most novel component, and it is quite intuitive why it would be helpful. It is not a pure RL approach. There is an initial SFT stage followed by PPO. The results across multiple relevant datasets show clear improvement over search-r1.

**Questions:**

How extensively did you attempt to get GRPO to improve over PPO? Does your comment about learned rewards imply that training was not stable? Was Pass@K sufficiently high to your liking? It is interesting to know whether we can expect models to automatically discover better trajectories, or we simply need to mimic the collected SFT data.

Could information gain be approximated by a judge w/ limited or no information of gold documents?

**Ethical Concerns:**

["NO or VERY MINOR ethics concerns only"]

**Final Justification:**

I think the paper has many strengths and is significant given that deep research is highly popular yet very few technical details have been published on how to create a deep research system. The concerns were minor, yet the author response addresses them nicely. I did not see any major concerns from others that convince me to change my score.

**Quality:**

3

**Strengths And Weaknesses:**

Strengths

1. This is one of the hottest topic areas, combining search and multi-step agentic rollouts. In particular, this adds an interesting reward in the form of information gain. There remains only limited research on RL + search + generation despite it's popularity.

2. The result shows a clear improvement over the next best method.

3. There is extensive ablation to help understand the importance of SFT, RL algo, rewards, etc.

Weaknesses

1. Perhaps would be more interesting to explore datasets with many gold retrieval labels. Although this seems alleviated by the Self dataset. The Self dataset could include more details, and it is not clear if it is as high quality as the other included benchmarks --- it is strange it is the one dataset where GRPO did not do as well given that the reward seems most aligned w/ the Self dataset.

2. SFT seems to be particularly critical. In other words, collecting more of this trajectory data could be greater value than algorithmic developments in the short term.

3. It could be interesting to evaluate deep research industry systems more directly, although would be understandably out of scope.

---

> ### Author Rebuttal · Authors · 2025-07-29
>
> We greatly appreciate reviewer **XiLb’s** recognition of the significance, clarity, and originality of our work. We are especially grateful for the positive comments regarding the introduction of information gain as a reward signal, the comprehensive ablation studies, and the clear improvements demonstrated over prior methods.
>
> Your recognition is greatly appreciated by our research team.
>
> Below, we address your questions and suggestions in detail.
>
> ------
>
> ## **1. Exploring Datasets with Gold Retrieval Labels & the Quality of the SELF Dataset**
>
> We fully agree that evaluating on datasets with many gold retrieval labels provides meaningful comparison. However, most widely-used QA benchmarks lack gold retrieval annotations, which motivated our construction of the **SELF** dataset.
>
> As described in Section 3.2, the SELF dataset was built by first manually browsing the web to collect authentic human browsing trajectories and identify key decision points. To scale up data collection while maintaining quality, we leveraged strong LLMs as autonomous agents to plan and make retrieval decisions, based on these initial human examples. We then performed strict quality control, filtering out queries that could be correctly answered without retrieval, or that could not be correctly answered even with all relevant information.
>
> *We will add more details about the construction and quality assurance of the SELF dataset in the revised version.*
>
> Regarding the performance of GRPO on the SELF dataset, although one might expect GRPO to excel given the reward signal alignment, we observed that PPO slightly outperformed GRPO, especially on SELF.
>
> We believe this is because our InForage model is first supervised-fine-tuned on constructed reasoning trajectories from SELF, giving PPO’s critic a well-informed starting point and leading to more accurate advantage estimation. In contrast, GRPO is less able to benefit from this supervised initialization, resulting in its slightly lower performance.
>
> ------
>
> ## **2. The Critical Role of SFT Data Collection**
>
> Thank you for highlighting this point. We agree that collecting high-quality trajectory data for SFT is particularly valuable at the current stage. High-quality trajectories provide rich, step-by-step supervision signals that closely mirror the actual reasoning and retrieval processes required by the target tasks. This, in turn, makes it easier for the model to receive rewards during the RL training stage, thereby alleviating the sparse reward issue.
>
> Nonetheless, we believe that both high-quality data and algorithmic advances are crucial and complementary.
>
> Rich supervision signals from data unlock the full potential of advanced algorithms, while improved algorithms help us better utilize and generalize from the available data.
>
> *We will further emphasize this interplay in the revised version.*
>
> ------
>
> ## **3. Direct Evaluation on Deep Research Industry Systems**
>
> We appreciate this suggestion.
>
> Direct evaluation on commercial deep research systems would indeed be valuable; however, current industrial systems are often closed-source, use non-aligned retrieval modules, or lack transparent implementation details, making direct comparison infeasible at this stage.
>
> ------
>
> ## **Responses to Specific Questions**
>
> ### **1. How extensively did you attempt to get GRPO to improve over PPO?**
>
> As noted, the SFT stage enables PPO’s critic to start from a well-informed value function, leading to more accurate advantage estimation and higher performance in our setting. We conducted extensive ablation, but GRPO was generally less able to benefit from this supervised initialization.
>
> ### **2. Does your comment about learned rewards imply that training was not stable?**
>
> We found that GRPO’s stability depends on the choice of group size and the reliability of verifiable reward signals. In our experiments, PPO’s critic structure contributed to more stable and effective training.
>
> ### **3. Was Pass@K sufficiently high to your liking?**
>
> We primarily report pass@1 accuracy, as aggregating multiple answers from different rollouts is non-trivial for QA tasks and may not reflect realistic usage.
>
> We consider pass@1 a more practical and fair metric.
>
> As shown in our results, InForage consistently outperforms baseline models.
>
> ### **4. Can models automatically discover better trajectories, or do they mainly mimic SFT data?**
>
> Thank you for raising this point. Our SFT data, synthesized from real web browsing records, provides high-quality supervision and is critical for model foundation. During RL post-training, the model is encouraged to autonomously explore and discover improved trajectories guided by reward signals. In practice, RL-based post-training further improves performance on top of SFT, suggesting the model can move beyond simply mimicking supervised data.
>
> ### **5. Could information gain be approximated by a judge without access to gold documents?**
>
> Information gain is a crucial signal, but many datasets lack intermediate labels. To address this, we consider two potential strategies:
>
> - **(1)** Using strong models with predefined workflows to simulate and recover valid intermediate steps, retaining those trajectories that ultimately lead to correct answers as additional training data;
> - **(2)** Employing a relevance model to estimate information gain via query-document relevance.
>
> However, the latter’s accuracy is highly dependent on the relevance model.
>
> *We will discuss these strategies and their limitations more thoroughly in the revised manuscript.*

---

> > ### Comment · Reviewer_XiLb · 2025-08-05
> >
> > Thank you for the response.
> >
> > There is a deep research API you can use. Although like I mentioned, it is out of scope. https://platform.openai.com/docs/guides/deep-research
> >
> > For Pass@K, perhaps I did not word the question directly. I am curious whether it was sufficient to have K sample generations per output. I am not sure what is the value of K you used (I checked the code, but could not tell based on the training script).
> >
> > I wonder if you can further substantiate that RL improve quality of rollouts beyond SFT. Sure, it generalizes better, but perhaps the rollout quality is still similar to the ground truth data. For example, I wonder if it naturally does more turns than in the SFT data.

---

> > > ### Author Response · Authors · 2025-08-05
> > >
> > > **Thank you for your thoughtful follow-up questions and suggestions. We address each point below:**
> > >
> > > ---
> > >
> > > ### 1. Evaluation with Deep Research API
> > >
> > > Per your suggestion, we evaluated the `o4-mini-deep-research` API on our self-constructed test set of 501 web-browsing–derived samples, which closely mirror real-world deep research scenarios where the model must gather evidence from the web.
> > >
> > > In our initial attempts, the deep-research API defaulted to generating long-form, report-like answers. To enable fair metric computation, we explicitly instructed the model to provide direct and concise responses. Even so, using the exact match (EM) metric, the deep-research API achieved an EM of 0.146 (73/501), which is notably lower than InForage.
> > >
> > > On closer inspection, we observed that many predicted answers included additional context or paraphrased the gold answer. For example:
> > >
> > > ```json
> > > {
> > >   "question": "What was the verdict in the case where the Australian soccer player known for her goal-scoring prowess, who pleaded not guilty to racially aggravated harassment, appeared at the British court located in a district sharing its name with a royal palace?",
> > >   "answer": "Not guilty",
> > >   "predicted_answer": "Sam Kerr was found not guilty of racially aggravated harassment at Kingston Crown Court. ([url1](url2))",
> > >   "em": 0,
> > > }
> > > ```
> > > In this case, although o4-mini-deep-research provided the correct answer, its additional explanations led to a mismatch with the expected answer string, resulting in a 0 EM score.
> > >
> > > To account for partial matches, we also computed an "inclusion-EM" score: if the predicted answer contained the gold answer string, we counted it as correct. This raised the EM for o4-mini-deep-research to **37.9%** (190/501), but it still lagged behind InForage at **44.1%** (221/501). We attribute InForage’s advantage to its in-domain training on the self-constructed dataset.
> > >
> > > ---
> > >
> > > ### 2. Clarification on Pass@K
> > >
> > > **Regarding the rollout number during training**:
> > >
> > > If your question refers to the rollout count used in training, this is set via the `actor_rollout_ref.rollout.n_agent` parameter in the script. For PPO training, we set this value to 1, as PPO utilizes a critic to estimate the advantage and during our expeirments, we find that rollout=1 is efficient and does not sacrifice performance. For GRPO (used in our ablation studies), which relies on multiple rollouts to estimate group-relative advantage, we set the rollout number to 8. Across these experiments, we found that GRPO generally underperformed compared to PPO, though it occasionally outperformed PPO on specific datasets. While our logs do not capture the number of valid or correct rollouts per sample, we observed a consistent increase in reward scores for both PPO and GRPO, indicating that the sampled rollouts are effectively guiding the optimization process.
> > >
> > > **Regarding the number of samples during evaluation**:
> > >
> > > If you are asking about the number of answer samples per evaluation, as noted in our previous response, we adopt a Pass@1 setting—meaning we sample only one answer trajectory for each evaluation example. This mirrors practical deployment scenarios, since aggregating multiple answers from different rollouts is non-trivial in open-ended QA and does not reflect realistic usage.
> > >
> > > ---
> > >
> > > ### 3. RL Improvement over SFT
> > >
> > > Regarding whether RL improves rollout quality beyond SFT:
> > >
> > > - The SFT-only model achieves an average score of **36.1**, providing a solid foundation.
> > > - In contrast, the model trained with RL alone reaches an average score of **34.0**, indicating that without SFT, the model is less able to discover effective trajectories for RL optimization. We attribute this to the complexity of our self-constructed dataset compared to previous datasets like NQ or HotpotQA, which makes valid trajectory exploration more challenging for a vanilla model.
> > > - After combining SFT and RL, the score increases to **41.2**, showing that, with a good foundation of SFT model, RL enables the model to find more effective strategies beyond the SFT data alone.
> > > - Training logs indicate the average number of valid search actions rises from 1.38 to 1.75 (over 300 steps) during SFT, and further from 1.76 to 2.41 (over 300 steps) during RL. This suggests RL enables deeper and more autonomous exploration compared to SFT alone.
> > > - In summary, RL not only improves answer quality but also leads the model to perform more reasoning turns than SFT.
> > >
> > > ---
> > >
> > > **We appreciate your suggestions and will add these clarifications and supporting results in our revised manuscript.**

---

### Author Response · Authors · 2025-08-08
**General Response**

Dear Reviewers, AC, SAC, and PC,

First, we would like to sincerely thank you for your thoughtful and constructive feedback. We greatly appreciate the time and effort invested in reviewing our work.

We are encouraged by the reviewers’ recognition of our paper’s novelty, technical soundness, and contributions to the field. In particular, we appreciate the acknowledgment of:


- The strong conceptual motivation and clear problem formulation underlying our work.
- The principled and innovative integration of multiple reward signals to drive model learning.
- The clarity and accessibility of our methodological presentation.
- The comprehensive and rigorous experimental validation.
- The overall significance, originality, and broader impact of our contributions to the research community.

We are grateful for the reviewers’ recognition of these strengths, and for their positive comments regarding the theoretical motivation, practical contributions, and ablation studies presented in our work.

During the rebuttal phase, we have addressed each of the reviewers’ concerns and questions in detail. We are pleased to note that **all reviewers have acknowledged our efforts in the rebuttal phase, confirmed that their concerns have been addressed, and offer a positive assessment of our work**. The feedback and suggestions provided have significantly strengthened our work.

In the final version, we will integrate the additional results, analyses, and clarifications provided in our rebuttal. Thank you again for your invaluable input and support.

---

### Decision · Program_Chairs · 2025-09-17

**Decision:**

Accept (spotlight)

**Comment:**

The paper proposes InForage, a reinforcement learning framework for search-enhanced reasoning grounded in Information Foraging Theory. It rewards intermediate retrieval quality (information gain/coverage) alongside final answer accuracy and an efficiency penalty, trained with human-guided web-browsing trajectories. Across general QA, multi-hop, and a new real-time web QA setting, InForage outperforms baselines.

### Strengths of the paper:
- The reviewers (XiLb) highlight the novelty of the information-gain reward and the paper’s timeliness for agentic, search-based RL; the proposed method made clear improvements over Search-R1 and it conducted extensive ablations dissecting SFT/RL/rewards.
- The work is technically sound with a well-designed three-component reward, thorough methodology (ablations/statistics), clear IFT motivation, and meaningful significance.
- The reviewers find the IFT grounding solid and well-justified, presentation clear, and experiments comprehensive across methods/datasets.

### Weaknesses of the paper:
- The reviewer sR8g requests larger-model scaling evidence (13B+) and additional baselines (e.g., FLARE, advanced IRCoT). The reviewer notes inconsistent benefits from the efficiency penalty and missing precision on key definitions (coverage C, “information scent”) and also worries about reliance on the self-constructed dataset.
- The reviewer XiLb wants datasets with gold retrieval labels, observes SFT is particularly critical, and suggests direct comparisons to industry deep-research systems.
- The reviewer ipjm asks for more detail on the information-gain reward (ideally a worked example) and a fuller discussion of limitations/complexities, including compute costs and debuggability.

### Primary reasons for Accept (Spotlight)
The primary reasons for recommending Accept (Spotlight) are that the paper makes a principled and timely contribution by operationalizing Information Foraging Theory into a tractable reinforcement learning framework for search-enhanced reasoning, introducing a mid-trajectory information-gain reward that is both novel and well-justified. Empirically, the method demonstrates consistent and robust improvements across multiple QA benchmarks and model scales, including outperforming strong baselines such as FLARE and showing competitive results against closed-source deep-research APIs. The authors further strengthened the submission during rebuttal by adding larger-model results (14B), expanded baseline comparisons, and clearer formalizations of the information-gain and coverage metrics.

### Summary of the discussion and rebuttal
The authors provided detailed and constructive responses to the reviewers’ concerns. For R-XiLb, they clarified the differences between PPO and GRPO, explained rollout settings, and showed that reinforcement learning leads to deeper and more effective search trajectories beyond SFT alone. For R-sR8g, the authors expanded their experiments by adding FLARE baselines at 3B/7B scales, reporting 14B model results, and clarifying the definitions of coverage and information scent, which addressed the reviewer’s main concerns and led to an upgraded score. For R-ipjm, they provided a worked example of the information-gain reward and clarified the nature of their browsing trajectories, while acknowledging limitations such as computational cost and dataset reliance. Overall, the rebuttal successfully addressed key issues raised by the reviewers, strengthening the paper’s clarity, empirical support, and theoretical framing.